On growth and form of irregular coiled-shell of a terrestrial snail: Plectostoma concinnum (Fulton, 1901) (Mollusca: Caenogastropoda: Diplommatinidae)

Liew Thor-Seng 1 2 3 thorsengliew@gmail.com
Kok Annebelle C.M. 1 2
Schilthuizen Menno 1 2 3
Urdy Severine 2 4 5
1 Institute Biology Leiden, Leiden University , Leiden , The Netherlands
2 Naturalis Biodiversity Center , Leiden , The Netherlands
3 Institute for Tropical Biology and Conservation, Universiti Malaysia Sabah , Jalan UMS, Kota Kinabalu, Sabah , Malaysia
4 Centrum Wiskunde & Informatica , Science Park, Amsterdam , The Netherlands
5 University of California San Francisco (UCSF), Anatomy Department, Genentech Hall , San Francisco, CA , United States
Jungers William
Electronic publication date: 2014 May 15
Publication date: 2014
Volume: 2
Electronic Location ID: e383
Received 2014 Mar 14; Accepted 2014 Apr 25
Copyright: © 2014 Liew et al.
Copyright year: 2014
Copyright holder: Liew et al.
License: This is an open access article distributed under the terms of the Creative Commons Attribution License, which permits unrestricted use, distribution, reproduction and adaptation in any medium and for any purpose provided that it is properly attributed. For attribution, the original author(s), title, publication source (PeerJ) and either DOI or URL of the article must be cited.
License URL: https://creativecommons.org/licenses/by/4.0/

Keywords: 3D morphometrics, Malaysia, Limestone hills, Open coiling, Borneo, Opisthostoma, Commarginal ribs, Radial ribs, Heteropmorph

Funding: Ministry of Higher Education, Malaysia and Universiti Malaysia Sabah Outbound Study Grant, LUSTRA, LUF Internationaal Studie Fonds, all from Leiden University Swiss National Science Foundation 200021_124784/1 PA00P3-136478 Netherlands Organisation for Scientific Research NWO, ALW 819.01.012 LTS received support from Ministry of Higher Education, Malaysia and Universiti Malaysia Sabah. ACMK received Outbound Study Grant, LUSTRA, and LUF Internationaal Studie Fonds, all from Leiden University. SU was supported by the Swiss National Science Foundation (200021_124784/1 and PA00P3-136478). This project is funded by Netherlands Organisation for Scientific Research (NWO, ALW 819.01.012). The funders had no role in study design, data collection and analysis, decision to publish, or preparation of the manuscript.

==============================
The molluscan shell can be viewed as a petrified representation of the organism’s ontogeny and thus can be used as a record of changes in form during growth. However, little empirical data is available on the actual growth and form of shells, as these are hard to quantify and examine simultaneously. To address these issues, we studied the growth and form of a land snail that has an irregularly coiled and heavily ornamented shell–Plectostoma concinnum. The growth data were collected in a natural growth experiment and the actual form changes of the aperture during shell ontogeny were quantified. We used an ontogeny axis that allows data of growth and form to be analysed simultaneously. Then, we examined the association between the growth and the form during three different whorl growing phases, namely, the regular coiled spire phase, the transitional constriction phase, and the distortedly-coiled tuba phase. In addition, we also explored the association between growth rate and the switching between whorl growing mode and rib growing mode. As a result, we show how the changes in the aperture ontogeny profiles in terms of aperture shape, size and growth trajectory, and the changes in growth rates, are associated with the different shell forms at different parts of the shell ontogeny. These associations suggest plausible constraints that underlie the three different shell ontogeny phases and the two different growth modes. We found that the mechanism behind the irregularly coiled-shell is the rotational changes of the animal’s body and mantle edge with respect to the previously secreted shell. Overall, we propose that future study should focus on the role of the mantle and the columellar muscular system in the determination of shell form.

Introduction

The physical form of organisms is central to different fields of biology, such as taxonomy, evolutionary biology, ecology and functional biology. Two major themes are the way the organism’s form changes as it grows and the way the organism’s form changes as it evolves. The formal investigation of growth and form was established by Thompson (1917) in his monumental On Growth and Form. In his book, Thompson studied the way organisms achieve their body form during growth, from the viewpoint of the mathematical and physical aspects of the ontogenetic processes. An extensively discussed example of these body forms are molluscan shells (see also Blake, 1878).

The molluscan shell, with the exception of those of bivalves, is a single structure that accommodates the animal’s soft body. The shell is secreted by the mantle edge, a soft elastic sheet of connective tissue covered by an epithelium (Wilbur, 1964). Accretionary growth occurs when the mantle lying inside the shell slightly extends beyond the current aperture and adds a shell increment to the margin (Wilbur, 1964). Thus, a shell is essentially a petrified ontogeny of the aperture (i.e., the mantle edge). A large amount of preserved ontogenetic information that can be used for Evo-Devo studies is available from both fossilized and extant shell-bearing species (Nützel, Lehnert & Fryda, 2006; Gerber, Neige & Eble, 2007; De Baets et al., 2012; Seuss et al., 2012; Urdy et al., 2013). In addition, the molluscan shell’s geometrically simple structure, resulting from a straightforward accretionary growth mode, makes it more popular than the body forms of other taxa in the study of theoretical morphospace (Dera et al., 2008). However, it remains challenging to empirically study the actual growth and form of a shell because of differences in the approaches of growth-orientated versus form-orientated studies (Ackerly, 1989; Okamoto, 1996; Rice, 1998).

There have been few changes in the study of shell growth rate since Wilbur & Owen (1964). The most commonly used method deals with the quantification of a shell’s linear dimensions such as shell length, shell width or number of whorl increments, which are then plotted against time (e.g., Kobayashi & Hadfield, 1996; Sulikowska-Drozd, 2011). Although these measurements are good estimators of the overall growth of the animal’s soft body (measured in weight; Oosterhoff, 1977; Chow, 1987; Elkarmi & Ismail, 2007; Silva, Molozzi & Callisto, 2010), they can hardly be linked with the accretionary growth process and spiral geometry of the shell. In addition, shell growth may be episodic because of different seasons, diurnal rhythms, or periods of activity and inactivity (Linsley & Javidpour, 1980). Thus, it is not easy to determine the temporal axis for shell ontogeny from a shell alone.

Similarly, studies of the changes in shell form throughout ontogeny tend to be based on the same morphometrics as in growth studies. These morphometrics are measured from the overall shell and are plotted against whorl or rib count, or rotation angle along the shell columella (Kohn & Riggs, 1975; Johnston, Tabachnick & Bookstein, 1991; Checa, 1991). However, these measurements do not provide an accurate record of shell form changes during ontogeny because the overall shell form is an accumulation of previous growth. Moreover, whorl count depends on a single imaginary coiling axis, which is missing in irregularly coiled shells (e.g., whorl-counting method by Verduin (1977)).

As a consequence, seldom are growth and form of a shell analysed simultaneously because the reference axes are usually not the same. For instance, a time axis may be used for shell growth, and a whorl count axis for shell form. Furthermore, these shell morphometrics do not closely approach the actual accretionary growth of the aperture in terms of form changes and growth trajectory changes (e.g., Ackerly, 1989; Stone, 1996; Stone, 1997; Rice, 1998).

Apart from the limitations in methodology, shell growth studies have initially been biased towards aquatic gastropods, and have mostly been conducted in the laboratory. For example, the chapter on molluscan growth in Wilbur & Owen (1964) mentions only a single shelled terrestrial gastropod species. Although the form and structure of aquatic and terrestrial gastropod shells are very similar, there are fundamental differences in the physiological and physical aspects of shell growth between them (Wagge, 1951; Kado, 1960; Fournié & Chétail, 1984). In recent decades, more studies on terrestrial gastropods have been conducted (e.g., Berry, 1962; Umiński, 1975; Oosterhoff, 1977; Baur, 1984; Ahmed & Raut, 1991; Johnson & Black, 1991; Kobayashi & Hadfield, 1996; Kramarenko & Popov, 1999; de Almeida & de Almeida Bessa, 2001a; de Almeida & de Almeida Bessa, 2001b; D’Avila & de Almeida Bessa, 2005; Bloch & Willig, 2009; Silva et al., 2009; Sulikowska-Drozd, 2011; Kuźnik-Kowalska et al., 2013; Silva et al., 2013). It is worth mentioning that most of these growth experiments used traditional morphometric methods and were conducted in the laboratory (but see Oosterhoff, 1977; Johnson & Black, 1991). Because discrepancies in growth patterns exist between field and laboratory experiments (Chow, 1987), further growth studies are needed from the natural habitat.

All the species investigated in the above-mentioned studies have shells that grow according to a regular coiling regime and with only simple radial ribs that are rather straight and do not protrude far from the shell surface, if any (but see Berry, 1962). For shells with irregular coiling, that is, those that pass through several dissociated growth stages, very little information is available as to how the growth and form changes during those different shell ontogeny phases. To alleviate all these limitations, we investigate the growth and form of an irregularly coiled and heavily ornamented tropical land snail species, Plectostoma concinnum, in its natural habitat.

We examined two aspects of shell growth and form: (1) the growth and form at three different whorl growing phases of the Plectostoma concinnum shell; (2) the switching between whorl growing mode and rib growing mode. First, we obtained a unified accretionary growth reference axis (hereafter termed “ontogeny axis”), namely the total arc length of the shell whorl (see “Definition of ontogeny axis” in Materials and Methods for more details), so that both shell growth and form data can be analysed together. Second, we obtained shell growth rate information that was measured as arc length of ontogeny axis (i.e., whorl length) added per day for live snails of different growth stages. Third, we quantified both the aperture form (size and shape), and the aperture growth trajectory (rotation, curvature and torsion) from a series of apertures (hereafter termed “aperture ontogeny profile”) that could be identified from the shells, by using 3D technology. Fourth, we explored the pattern of switching between whorl growing mode and rib growing mode that determined the number of ribs on the shell (see “Organisms” in Materials and Methods). Finally, we examined the associations between the growth and the form of the Plectostoma concinnum shell in all three whorl growing phases and both growing modes, from developmental-biological and theoretical-morphological points of view.

Materials and Methods

Ethics statement

The permissions for the work in the study sites were given by the Wildlife Department of Sabah (JHL.600-6/1 JLD.6, JHL.6000.6/1/2 JLD.8) and the Economic Planning Unit, Malaysia (UPE: 40/200/19/2524).

Organisms

The tropical terrestrial micromollusc subgenus Plectostoma consists of 79 species that are only known from limestone hills of Southeast Asia (Vermeulen, 1994; Liew et al., 2014). It is one of the most diverse genera in the Gastropoda in terms of shell form. In this study, we selected Plectostoma concinnum (Fulton, 1901), an endemic species in northern Borneo. This species is exclusively found in limestone habitat and thus presumably not limited by calcium availability. It occurs in high population densities with several millions of individuals estimated to live on limestone hills of less than 0.5 km2 (Schilthuizen et al., 2003).

In this study, we followed the terminology of Vermeulen (1994) in the discussion of the shell form of this species, and we used the term whorl growing mode and rib growing mode in the discussion of two different growth modes (e.g., Bucher, 1997; Hammer & Bucher, 2005). At least in the case of this particular species, we think these two terms are more precise than generic terms such as spiral and radial growth (e.g., Spight & Lyons, 1974; Vermeij, 1980). For the whorl growing mode, three growth phases can be distinguished, namely, spire, constriction and tuba.

As an adult, the species has about 5.5–6.5 shell whorls and is about 3 mm in height and 3.5 mm in width. The protoconch is smooth (Fig. 1A). The first 5 or 6 whorls of the teleoconch are regularly coiled (hereafter termed “spire”) while the last half whorl (hereafter termed “tuba”) is detached from the spire (Fig. 1A). The transition from the spire to the tuba is marked by a narrowing of the whorl (hereafter termed “constriction”), where calcareous lamellae are formed inside the aperture (hereafter termed “constriction teeth”) (Figs. 1A, 3E and 3F). The three parts are formed during the whorl growing mode. It has an operculum which rests behind the constriction teeth when the animal’s soft parts withdraw into the shell (hereafter “the animal” refers to the foot, the columellar muscle, and the mantle). Such an extreme morphological transition between spire and tuba is also known in several other extant and fossil mollusk species (e.g., for gastropods: Savazzi, 1996; Vermeulen, 1994; Clements et al., 2008; Frýda & Ferrová, 2011; and for ammonoids: Okamoto, 1988; Okamoto, 1996). The shell growth of this species is definite and the whorl growing mode ends with a “differentiated” peristome.

Figure 1 Terminology used for Plectostoma concinnum in this study.

(A) Terminology used in the descriptions of shell, (B) terminology used in the descriptions of animal, (C) an example of a shell with a nail polish mark and with the spiral striation on the shell indicated, (D) marking scheme for a shell at rib growing mode, (E) marking scheme for a shell at whorl growing mode, (F) whorl length measured from a specimen and the spire part that attaches to tuba, (G) ontogeny axis consists of a concatenation of whorl lengths of a shell, and (H) tracing aperture outlines from a shell.

The shell exhibits regularly spaced projected commarginal ribs. As there is no standardisation in the rib morphology terminology, to avoid confusion, we use the term commarginal ribs (sensu Seilacher, 1991) for the type of ribs of Plectostoma concinnum because it describes the ribs with reference to ontogeny and form and thus is more accurate than other terminologies (such as “radial ribs” or “growth halt” sensu Laxton, 1970). These commarginal ribs are the product of a rib growing mode, which is entered when the animal’s mantle edge expands dramatically and forms an aperture that is much larger than the previous aperture produced in whorl growing mode. After shell deposition stops at this rib growing mode, the subsequent whorl growing mode continues from the aperture that was produced in the previous whorl growing mode. The switching between these two growing modes produces the projected commarginal ribs.

Definition of ontogeny axis

To analyse the growth rate in terms of ontogeny axis growth per day and the form changes in terms of aperture ontogeny profile over time, one needs to extract a set of homologous points in an ontogenetic series that reflect the accretionary spiral growth. These points have to be homologous in a biological sense meaning that the different growth stages of the same individual as well as those of several different individuals are comparable. These landmarks can correspond to the localisation of a specific structure (geometrical homology), to the temporal repetition of the same structure (serial homology) or to the occurrence of a developmental event such as the onset of metamorphosis or senescence (developmental homology) (Johnston, Tabachnick & Bookstein, 1991).

In Plectostoma concinnum, the spiral line at the anterior point of the aperture (Figs. 1C, 1G and 2A) fulfils the conditions for geometrical homology since such striations are produced by particular cells at the mantle edge (Salas et al., 2012). It corresponds to the point of the aperture with maximum growth rate and the curvature is maximal at this point (Figs. 1F, 1G and 2A). The successive protruded radial ribs fulfil the conditions for serial homology, while the protoconch-teleoconch boundary and the spire-tuba constriction define developmentally homologous events. Thus, we used an ontogeny axis, starting from the protoconch-teleoconch boundary (Figs. 1F and 1G), and obtained by concatenating the arc lengths measured form the points of maximum growth rate between successive protruded radial ribs. Our ontogeny axis is similar to those used by Gould (1969), Vermeij (1980), Savazzi (1985), Savazzi (1990), Checa (1991) and Johnston, Tabachnick & Bookstein (1991). The ontogeny axis of each shell was obtained and the growth and from variables derived below were then plotted and analysed along this ontogeny axis. Different positions along the ontogeny axis represent different growth stages of a shell.

Figure 2 Steps in the analysis of aperture (i.e., animal) orientation changes.

(A) Segmentation: each segment consists of whorl and rib part. For each analysis, two segments were included which represent the two animal orientations, namely, the newly formed segment (NEW—in yellow) and the previously formed segment (OLD—in red segment), (B) reset the NEW segment orientation according to the animal axes, (C) translation: move the OLD segment to NEW segment, so that the anterior points of the two segments were aligned, (D) rotation of OLD segment around x-axis corresponding animal left or right tilting from animal’s anterior view, (E) rotation of OLD segment around y-axis corresponding to rotation of the dorsoventral axis (shell growth direction), and (F) rotation of OLD segment around z-axis corresponding to rotation of the animal clockwise or anticlockwise rotation from animal’s dorsal view. Scale bar, 0.5 mm.

Figure 3 Animal orientations and formation of constriction teeth of Plectostoma concinnum at different growth phases.

(A)–(C) Orientation of animal with respect to shell at spire phase, tuba phase, and adult. (D) Constriction teeth begin to form inside the shell at the end of spire growth. (E)–(F) Constriction teeth become more prominent during the tuba growth.

Experimental design and sampling

The growth experiments were carried out at two limestone outcrops in the vicinity of Kampung (Village) Sukau, Lower Kinabatangan Valley in the state of Sabah, Malaysia, between 20 April and 10 May, 2011. These two isolated limestone outcrops, Batu Kampung (5°32′11″N 118°12′47″E) and Batu Pangi (5°31′59″N 118°18′44″E), are located 10 km apart, and thus climatic differences are negligible. Thanks to the rainy season, the microclimates were constant throughout the three weeks of the experiment (File S1). Six rock surfaces (ca. 10 m2 each, hereafter referred to as “plots”) with high densities of Plectostoma concinnum and similar ecological conditions, in terms of moss and lichen coverage, and shadiness, were selected. The numbers of replicated plots, growth experiment durations and specimens examined are shown in Table 1.

Table 1 Experimental setups and number of specimens used in this study.

Dataset	Hill	Plots	Duration and date of experiment	Number of specimens	
1	Kampung	1	2 days (7 May–9 May 2011)	18	
2	Kampung	2	3 days (7 May–10 May 2011)	11	
3	Pangi	1	13 days (20 April–3 May 2011)	6	
4	Pangi	2	13 days (20 April–3 May 2011)	3	
5	Pangi	3	11 days (22 April–3 May 2011)	12	
6	Pangi	4	4 days (4 May–8 May 2011)	15	

We used a capture-mark-recapture method (CMR) in the plots (e.g., Cain, Cook & Currey, 1990). In each one-hour session, we collected between 100 and 200 juveniles of Plectostoma at different growth stages. Then, in a field lab, using a dissecting microscope, we marked each shell with a nail polish mark located on either the second most recently grown rib (if the snail was at rib growing mode) (Fig. 1D) or the most recently grown rib (if the snail was at whorl growing mode) (Fig. 1E). We used this marking scheme instead of one in which a mark was placed on the aperture edge, to prevent the nail polish to come in direct contact with the animal mantle. Our nail polish marking technique fulfilled the general requirements for CMR approach (sensu Henry & Jarne, 2007). The marks were clearly visible, persisted for at least two months under field conditions and had no noticeable effect on the mantle edge. All marked individuals were released at their exact point of capture within 24 h and were recaptured between 2 and 13 days later (see Table 1). All recaptured individuals were killed by drying. A total of 97 shells were thus obtained from both study sites, of which 15 had suffered aperture damage and were discarded. All specimens were deposited as voucher samples in the BORNEENSIS collection, Universiti Malaysia Sabah—BOR.

The remaining 82 shells (65 juveniles and 17 fully grown at the time of recapture) were used for the following analyses. For shell growth rate analysis (Part 1), we used the 65 juvenile shells (36 from Batu Pangi; collection sample BOR 5653 and 29 from Batu Kampung; collection sample BOR 5654). For the aperture profile analysis (Part 2), we quantified (a) aperture shape and size for five representative shells (out of the 65 juvenile shells) at different growth stages; and (b) growth trajectory of a fully grown shell (out of the 17 adult shells). For the analysis of whorl and rib growing mode (Part 3), we examined (a) the number of switches between the two growing modes in the 17 fully grown shells that collected from the same location (collection sample BOR 5652); and (b) the pattern of whorl spacings between two rib growing modes of the 35 shells (out of the 65 juvenile shells) that had grown beyond the constriction.

Part 1—Shell whorl arc length growth rate along the shell ontogeny

Each of the 65 juvenile shells was photographed (with a Leica DFC495 attached to a Leica M205C microscope). Photographs were taken in apical view (Fig. 1F and File S2). For those specimens that grew up to the tuba stage, we aligned the tuba with a plane and we took additional photographs (File S2). The arc length at the point of maximum growth rate was calculated using the program Leica Application Suite V3.7.0. Although the arc length is measured from two-dimensional images (Fig. 1F), it is a good proxy for the three-dimensional arc length (Fig. 1G and File S3: r = 0.82, n = 251 (3 shells), p = 0.000). We thus obtained 5,475 arc lengths measured between successive ribs and pooled these data (File S4). The arc length of the ontogeny axis for each of the 65 shells was calculated as the sum of all the arc lengths between successive ribs of each shell.

Based on the nail polish mark on the shell, we measured the arc length before and after the growth experiment. Then, we calculated growth rate as the whorl arc length (i.e., ontogeny axis) added over the duration of the experiment (i.e., mm day−1). We tested for the correlation between the measured growth rates and the position of the specimen on the ontogeny axis prior to the growth experiment. The analyses were done separately on the two growth phases of Plectostoma concinnum, namely, spire and tuba. Spearman correlation was used since the data were not normally distributed.

Part 2—Aperture ontogeny profile changes between spire growth phase and tuba growth phase

In this part, we examined the animal’s orientation and aperture form changes along the ontogeny axis. First (Part 2a), we obtained aperture forms by quantifying the traced aperture on 3D shell models. Second (Part 2b), we quantified aperture growth trajectory changes by examining the animal orientation with respect to its shell and by quantifying the spiral geometry of the ontogeny axis in terms of curvature and torsion estimators.

We used microcomputed X-ray tomography to obtain 3D models of the various growth stages of P. concinnum (n = 6). Five of these 3D models (immature shells) were used for aperture outline analysis while one 3D model of an adult shell was used for animal rotation analysis (see below). The microcomputed tomography used a high-resolution micro-CT scanner (SkyScan, model 1172, Aartselaar, Belgium). The scan conditions were as follows: voltage—100 kV; pixel—1336 rows × 2000 columns; camera binning—2 × 2; image pixel size—3.42 µm; rotation step—0.4°; and rotation—360°. Next, the volume reconstruction on the acquired images was performed with the manufacturer’s software NRecon ver. 1.6.6.0 (SkyScan). The images were aligned to the reference scan and reconstruction was done with the following settings: beam hardening correction—100%; reconstruction angular range—360°; image conversion (dynamic range)—ca. 0.12 and ca. 20.0; and result file type—BMP. Finally, 3D models were created from the reconstruction images with the manufacturer’s software CT Analyser ver. 1.12.0.0 (SkyScan) with the following settings: binary image index—1 to 255; and saved as digital polygon mesh objects (*.ply format). The 3D models were then simplified by quadric edge collapse decimation to ca. 30,000 faces, with a method implemented in MeshLab v1.3.0 (Cignoni, Corsini & Ranzuglia, 2008). The subsequent analyses for the digital 3D shell models were done in 3D modelling open source software—Blender ver. 2.63 (Blender Foundation, www.blender.org).

Part 2 (a) Aperture form changes between spire growth phase and tuba growth phase

The acquisition of aperture outlines and their trajectories was done in Blender software with its embedded object-oriented programming language Python. We wrote custom Python scripts to extract the outline points’ coordinates for shape analysis (File S5). We used the “grease Pencil” tool of Blender to trace the aperture and ontogeny axis outlines on the five immature shell 3D models (Figs. 1G and 1H). Then, we converted these traced outlines into Bezier curves, where the outlines were represented by a series of points with three-dimensional Cartesian coordinates. We obtained outline data of five 3D shell models with a total of 33 apertures (File S6), which were then analysed together with their homologous ontogeny axis.

We obtained the aperture outline perimeter by summing the distances between the successive points of each aperture outline. Before that, we smoothed each of the Bezier curve outlines by a three-dimensional Elliptic Fourier Analysis (hereafter termed “3D EFA”; Kuhl & Giardina, 1982; Godefroy et al., 2012) to minimize the possible noise coming from the digitalization process. We ran the 3D EFA with the following parameterization: number of harmonics = 5, starting point = anterior point, and outline orientation = clockwise. We used five harmonics because they were sufficient to reliably describe the aperture outlines (File S7). Next, we reverted 3D EFA function so that each outline was reconstructed from the same set of five harmonics and by using 100 sample points along the outline. Finally, we extracted the aperture perimeters from the 100 points of each outline.

We obtained principal component analysis (PCA) scores from normalized coefficients of 3D EFA for each of the 33 aperture outlines (hereafter termed “shape scores”). The coefficients of the 3D EFA harmonics were normalized according to Godefroy et al. (2012) so that they were invariant to size and rotation. After normalization, all of the 30 normalized Fourier coefficients for each of the 33 aperture outlines were analysed by PCA in R statistical package 2.15.1 (R Core Team, 2012). R scripts are in File S4.

The aperture perimeter and shape scores of each aperture were examined together along the ontogeny axis. In addition, a linear regression was performed on the spire aperture perimeter changes along the ontogeny axis in R statistical package 2.15.1 (R Core Team, 2012).

Part 2 (b) Aperture growth trajectory changes between spire growth phase and tuba growth phase

In the plots at Batu Kampung, we collected additional specimens at different growth stages to examine the growth trajectory of the aperture and the orientation of the living animal with respect to its shell. The living individuals were carefully picked up with a pair of soft forceps while active, were immediately frozen with Freeze spray (KÄLTE, Art. Nr. 20.844.6.09.12.01) and preserved in 70% ethanol. The body rotation of the animals of different shell growth stages was examined with scanning electron microscopy (SEM).

We found that the highest projected point of the commarginal rib corresponds to the anteroposterior axis of the animal. In addition, the changes in the orientation of successive segments correspond to the changes in orientation of the animal as evidenced by the homologous anterior landmark of the aperture (Figs. 3A, 3B and 3C). Hence, the growth trajectory changes in terms of animal rotation can be inferred directly from the shell. We therefore quantified the orientation changes of the animal along the shell ontogeny from a 3D model of an adult shell with Blender. We restrict this analysis to the ontogeny corresponding to the 1.5 whorls before the constriction up to maturity where the most drastic changes in shell coiling direction occur (File S8).

We obtained the ontogeny axis for the shell and then separated the digital 3D shell into segments corresponding to successive commarginal ribs (Figs. 2A and 2B, File S8). We obtained changes in the rotation between two consecutive segments (hereafter termed: “NEW” and “OLD” segments). The changes in animal orientations were inferred from these two segments with respect to the anatomical directions of the animal. First, we aligned the anteroposterior axis of the NEW shell segment with the x-axis of the global 3D Cartesian system (Fig. 2B). Second, we aligned the anterior point of the OLD segment to the anterior point of the NEW segment (Fig. 2C). Third, we rotated the OLD segment along the x, y and z axes until it was aligned with the NEW segment (Figs. 2D, 2E and 2F). Finally, the rotation changes (in angles) were plotted along the ontogeny axis.

The rotations around the three animal anatomical directions were interpreted as following. First, rotation around x-axis corresponds to the aperture “inclination” with respect to the previous aperture (Fig. 2D). It corresponds to the direction where the animal tilts to right or left. Second, the rotation around y-axis corresponds to the rotation of the dorsoventral axis (i.e., shell growth direction). Third, the rotation around z-axis corresponds to the rotation of the anteroposterior axis (rotation of the aperture plane around its centroid). When the animal is viewed in dorsal view, we describe this rotation as either clockwise or anticlockwise. From our observation (see above), it seems the most important changes in animal orientation at different growth phases are the rotations around the x-axis and the z-axis (Figs. 2B, 3A, 3B and 3C).

We are aware that the discrete rotation analysis between shell segments may not realistically reflect the continuous changes of the growth trajectory. Thus, we estimated curvature and torsion, two parameters that are convenient to describe a 3D spiral (Okamoto, 1988; Harary & Tal, 2011). These were estimated from the same adult shell as above (File S9). The curvature (Kappa) and torsion (Tau) were estimated from each sample point along the ontogeny axis by a weighted least-squares fitting and local arc length approximation (Lewiner et al., 2005). The calculation was done by custom written Python scripts, which were run in the Blender environment (File S5). The estimation was done with 100 points on the left and right sides for each sample point. The value of curvature is a positive value; the ontogeny axis is a straight line (i.e., shell is an orthocone) when Kappa = 0; and the larger the curvature, the smaller the radius of curvature (1/Kappa). The torsion Tau estimates the deviation of the curve from a plane—the zero value meaning that the shell is planispiral. In addition, a negative/positive torsion value corresponds to a left-handed/right-handed coiling respectively.

Part 3—Switching between whorl growing mode and rib growing mode; frequency and trend

We examined the variation of the number of ribs, which indicates the number of switches between the two growing modes. Then we compared the switching patterns among shells varying in rib number.

Part 3 (a) Variation of total number of ribs between shells

The numbers of ribs were counted on the spire and tuba parts of each of the 17 adult shells which had completed their shell growth under similar ecological conditions in our field experiment. Because the number of ribs on the spire correlated with the number of ribs on the tuba (see Results), in a subsequent analysis, we counted the number of ribs and arc length of the ontogeny axis on the fully grown spire of 35 juvenile shells. We tested if there is a correlation between the total number of ribs and the total ontogeny axis length. As all data were normally distributed, we used Pearson correlation in R 2.15.1 (R Core Team, 2012); R scripts may be found in File S4.

Part 3 (b) Switching trends between the whorl growing mode and the rib growing mode

We plotted 3,263 arc lengths (both spire and tuba) between successive ribs of the 35 shells along the same ontogeny axis.

Results

Part 1—Shell whorl arc length growth rate along the shell ontogeny

The growth rates are measured in mm/day along the arc length travelled by the point of maximal growth rate during ontogeny (n = 65, File S4). The absolute shell whorl arc lengths added to the shells during the growth experiments are found in File S10.

Figure 4 shows the growth rate variations along the ontogeny axis for the spire and tuba growth phases of 65 shells. For the growth patterns of the spire, the growth rate is positively correlated with the ontogeny axis (r = 0.45, n = 30, p = 0.01). On the other hand, after the constriction, the growth rate is negatively correlated with the ontogeny axis (r = −0.38, n = 35, p = 0.02). These data suggest that O. concinnum may follow a S-shaped growth curve (with regard to time), with the maximum growth rate occurring during the transitional phase (inflexion point).

Figure 4 Growth of shell whorl arc length along the shell ontogeny for 65 specimens.

Growth rate increases along the shell ontogeny for the spire part but decreases in the tuba part of the shell.

Part 2—Aperture ontogeny profile changes between spire growth phase and tuba growth phase

Part 2 (a) Aperture form changes between spire growth phase and tuba growth phase

Figure 5A shows the changes of aperture perimeter from around 5 mm until the end of the ontogeny axis. The aperture perimeter changes along the ontogeny of the five different specimens share a common trend. The perimeter of the aperture increases linearly, in an, on average, constant rate (ß = 0.166), between 5 mm and ca. 11 mm at the ontogeny axis (linear regression model: (aperture perimeter) = 0.166 (position of ontogeny axis) + 0.457, R2 = 0.97, F = 591.4, df = 1, 20, p = 0.000). Then, the aperture size decreases during the constriction part of the ontogeny before the size increases again during the tuba part of the ontogeny. It is important to note that this linear pattern and the constant rate of the aperture size increment were obtained from only five shells.

Figure 5 Aperture form changes along shell ontogeny axis.

(A) The apertures perimeter changes in the five specimens show unified patterns along ontogeny axis. (B) Changes of aperture shape (summarized in PC 1 scores, as measured from five specimens) along the ontogeny axis. Arrow points to the anterior direction of apertures. The part of the aperture that attaches to previous whorl (red line) and to subsequent whorl (blue line).

For the aperture shape analysis, the PCA reveals that the first three components accounted for 53.8%, 14.2%, and 9.7% of the total shape variation of all five sets of harmonics (File S11). The correlation analysis reveals that the first component is significantly correlated with 15 out of the 30 normalized Fourier coefficients, especially the Fourier coefficients of the first harmonics (File S11). Thus, we retained the PCA first component’s scores as shape descriptor of aperture (due to the nature of the EFA, the first harmonic contains a large part of the variation and most of the shape information; Kuhl & Giardina, 1982).

Figure 5B shows the changes of aperture shape along the ontogeny axis. During the spire part of the ontogeny, the aperture has a diamond shape with a round corner. Its perimeter is slightly convex at the right anterior, left anterior and posterior sides, but slightly concave at the right posterior side. Approaching the constriction part of the ontogeny, the diamond-shaped aperture becomes elongated along the anteroposterior axis with slightly rounded corners. At the tuba part of the ontogeny, the aperture has an ovate shape that is symmetrical along the anteroposterior axis, acute at the anterior and wide at the posterior.

Part 2 (b)—Aperture growth trajectory changes between spire growth phase and tuba growth phase

Figure 6 shows the rotational changes of each new segment with respect to the previous segment. Rotation around the x-axis at the constriction and part of the last whorl shows that the changes in the animal’s orientation are in the opposite direction compared to most of the spire and tuba parts of the ontogeny. There is no change of rotation direction around the y-axis as the shell follows a spiral growth. The magnitude of rotation in the y-axis is related to the whorl length between two ribs (confer Fig. 8). Rotations around the z-axis reveal that the rotational changes between two ribs for the spire and the tuba part of the ontogeny are in opposite direction.

Figure 6 Changes of an animal’s orientation in terms of standardised rotation in angle during the growth between two consecutive segments along the ontogeny axis.

(A) Rotational changes around x-axis—animal tilts to either left (negative angles) or right (positive angles), (B) rotational changes around y-axis—shell growth direction, and (C) rotational changes around z-axis—animal rotates either clockwise (negative angles) or anticlockwise (positive angles).

Figure 7 shows how the curvature and torsion values change along the ontogeny axis. The curvature value decreases rather constantly from ca. 3 to ca. 1 with small fluctuations along the spire part of the ontogeny. However, for the constriction to the tuba part of the ontogeny, the curvature value fluctuates between 0.9 and 1.3. Torsion values along the spire decrease gradually from 0.9 to 0.1. From the constriction onwards, however, torsion fluctuates wildly, becoming strongly negative before returning to positive values.

Figure 7 Curvature and torsion of a shell along the ontogeny axis.

(A) Curvature, inset shows curvature changes along the growth trajectory, (B) torsion, inset shows torsion changes along growth trajectory.

Part 3—Switching between whorl growing mode and rib growing mode; frequency and trend

Part 3 (a) Variation of total number of ribs between shells

The arc lengths measured between two consecutive spines in 35 individuals of Plectostoma concinnum were pooled together (3263 arc lengths in total, raw data in Supporting Information). There is no significant correlation between the number of spines and the total arc length (Pearson correlation, r = −0.22, n = 35, p = 0.2), highlighting that the number of spines varies extensively among individuals exhibiting a similar total arc length. However, there is a significant correlation between the number of spines before the constriction and the number of spines after the constriction (Pearson correlation, r = 0.55, n = 17, p = 0.02). This means that there is still a consistent ontogenetic pattern in this set of pooled data: the individual ontogenies do not vary to the extent that the spiral and tuba phase are mixed together in the pooled data.

Part 3 (b) Switching trends between the whorl growing mode and the rib growing mode

Figure 8 shows that the spacing between successive ribs increases constantly from right after the protoconch (i.e., at position 0) to ca. 8 mm along the ontogeny axis. The spacing between ribs then decreases until ca. 10 mm on the ontogeny axis (Figs. 1F and 8). Then, this spacing increases from ca. 10 to ca. 13 mm on the ontogeny axis, when the shell is about to form the constriction part. The spacing then decreases during the transitional constriction phase (from ca. 13 to ca. 14 mm on the ontogeny axis) and remains approximately constant during the tuba phase (from ca. 14 mm to the end of the ontogeny axis). Shells with different numbers of ribs show the same trend but of a different magnitude—the average rib spacing of densely ribbed shells being shorter than that of sparsely ribbed shells at the same growth stage.

Figure 8 Whorl arc length between two commarginal ribs.

Trends in whorl arc length between two commarginal ribs in 35 shells which vary in the number of ribs on the spire along the ontogeny axis.

Discussion

Growth and form of whorl growing mode in terms of aperture form and growth rate

The overall shell ontogeny of Plectostoma concinnum does not comply at all with the ideal shell growth model in which the growth parameters remain constant throughout the ontogeny. Although such ideal shell growth has been an essential part in the development of gastropod theoretical morphology (Moseley, 1838; Thompson, 1917; Raup, 1966), the shells of most gastropods do deviate to some extent (Blake, 1878; Raup, 1966; Gould, 1968; Vermeij, 1980; Urdy et al., 2010). The shell ontogeny of P. concinnum begins with a regular growth phase that approximates a dextral isometric logarithmic spiral (spire phase, between 0 and ca. 13 mm on the ontogeny axis), followed by a more variable transitional growth phase (constriction phase, ca. 13-ca. 14 mm of ontogeny axis), which gives way to an open-coiling growth phase (tuba phase, from ca. 14 mm to the end of the ontogeny axis). Thus, it provides a unique opportunity for us to investigate how shell form changes in relation to the growth rate.

Spire

The spire is dextral, has a regular growth trajectory and form, and thus its curvature and torsion estimators obey the 3D logarithmic spiral geometry with minor deviation (Fig. 7). During the growth of the spire, the aperture ontogeny profiles either remain the same or change in a constant manner. The aperture remains of almost the same shape (Fig. 5B), the aperture perimeter increases linearly and constantly, the animal (i.e., the mantle) always rotates clockwise (Fig. 6C) from the animal’s dorsal view (e.g. Figs. 2B and 2F), and the aperture inclination declines (Fig. 6A). These variables alter when the spire phase changes over to the constriction phase.

Constriction

The constriction part of the ontogeny breaks the simple logarithmic spiral growth rule. Every aspect of the aperture ontogeny profiles changes: the aperture shape differs from the spire aperture (Fig. 5B); the aperture perimeter drops, the animal (and its mantle edge) begin to rotate anticlockwise (Fig. 6C) from animal’s dorsal view (e.g. Figs. 2B and 2F), and the aperture inclination increases (Fig. 6A).

Our data show that changes in the animal’s orientation are responsible for the break in the preceding growth rule (Figs. 6A, 6C and 7). It has been shown theoretically that the rotation of the animal within the shell—which is equivalent to changing the pattern of growth rates around the aperture—is the cause behind the drastic changes in the coiling pattern that are observed in irregularly coiled ammonites (Okamoto, 1988; Okamoto, 1996) and cemented gastropods exhibiting distorted coiling (Vermeij, 1993; Rice, 1998). Our data support this hypothesis, and suggest that the deviation is caused by the continuous rotation of the mantle edge in the opposite direction to that of the spire part, during the accretionary growth process at the aperture.

Several studies have pointed out a general correspondence between the life position and the shell morphology in recent gastropods (Linsley, 1977; Linsley, 1978; Morita, 1991a; Morita, 1991b; Morita, 1993; Morita, 2003; Checa, Jiménez-Jiménez & Rivas, 1998; Vermeij, 2002), indicating that the life position of gastropods is almost equal to the gravitationally stable position of their empty shells. These studies argued that the direction and degree of coiling, as well as aperture shape are at least partly determined by the columellar muscle, the animal’s living position (at the time of shell secretion), and the previous whorl (‘road-holding’, Hutchinson, 1989; Checa, Jiménez-Jiménez & Rivas, 1998). Although some details are available regarding the structure and retraction function of the columellar muscle (Brown & Trueman, 1982; Kier, 1988; Frescura & Hodson, 1992; Thompson, Lowe & Kier, 1998; Suvorov, 2002), how the columellar muscle may act to affect shell morphogenesis is unknown.

In addition to the aperture shape and growth trajectory changes at the constriction phase, the aperture size also decreases along the shell ontogeny before increasing again when approaching the tuba phase. This process produces a narrower shell whorl, and is unlikely to be directly involved in the aperture rotation. Yet the constricted whorl might play a key role in the ontogeny of the tuba part of shell. At the beginning of the tuba phase, several constriction teeth are formed inside the constricted whorl. These constriction teeth are associated with the columellar muscle and thus could play a role in controlling the animal’s orientation with respect to the shell (Figs. 1B, 3D, 3E and 3F). For example, the columellar muscle could coil around the shell columella during the spire phase whereas the columellar muscle has to extend far from the shell columella during the tuba phase. Hence, the constriction teeth could serve as a holdfast for the columellar muscle and prevent it from shifting position while under tension (e.g., Signor & Kat, 1984; Price, 2003). This kind of internal structure is also common in other irregularly coiled gastropods (Savazzi, 1996). So, we support Suvorov’s view that the constriction teeth could be important for an effective management of shell orientation (Suvorov, 1993; Suvorov, 1999a; Suvorov, 1999b; Suvorov, 2002). The constriction phase might therefore pave the way for the later tuba phase in forming a detached whorl.

Tuba

Two significant aspects of the tuba part of the shell ontogeny are its anticlockwise rotation on the growth trajectory (Fig. 6C) and its detachment from the spire of the shell. At this phase, the aperture shape changes rapidly (Fig. 5B) and the aperture perimeter increases again with a trend similar to the spire (Fig. 5A). Here, we can show that the transition from a tightly-coiled and almost isometric dextral shell to an open-coiled tube only requires a relatively small and continuous change in the main growth direction. This is achieved by the continuous rotation of the animal within the shell, in the opposite direction as compared to the spire phase, possibly controlled by the columellar muscle as discussed above.

The continuous rotation of the aperture causes the later part of the tuba to detach from the spire. In the spire part, only a thin layer of shell is deposited at the right lateral part of the aperture along the surface of the previous whorl, causing fusion with the previous whorl. In contrast, during the tuba part of the ontogeny where the whorls are detached, thicker shell layers are deposited all around the aperture.

The shell whorl overlapping at the spire part is a more economical growth strategy than detached whorls (Heath, 1985; Stone, 1999; Stone, 2004). We suggest that the differences in growth rate between spire and tuba might result from the detached growth of the tuba. As the aperture size of the tuba and the later part of the spire is similar, and calcium is not a limiting resource for this limestone-dwelling species, the formation of the detached whorl may slow down because more time is required for its formation compared to the spire. However, as growth is determinate in this species, we expect growth to slow anyway at the onset of maturity with the development of the reproductive organs (e.g., Terhivuo, 1978; Lazaridou-Dimitriadadou, 1995).

Finally, the change from tight to open coiling in Plectostoma concinnum could provide an opportunity to revisit theoretical models on whorl overlap—the road-holding model (Hutchinson, 1989) and its mechanical effect on aperture shape (Morita, 1991a; Morita, 1991b; Morita, 1993; Morita, 2003). In his morphogenetic model, Morita (1991a) defines the mantle as a whole as a hydroskeleton which is usually in a state of expansion resulting from internal haemolymph pressure. Consequently, the mantle is simulated as a double elastic membrane connected by internal springs. Its physical state is supposed to be in balance between its internal stress and the forces acting on it, such as the pressure of the haemolymph, the pressure induced by the foot/columellar muscle and the boundary of the shell. The deformation of the mantle is then deduced from its stress field using a finite element analysis. Morita investigates the effect of a zone where the mantle cannot deform—presumably because of the foot/muscle/soft parts pressing on the mantle edge. He shows that initially circular walls change into elliptically elongated ones with pressure rising. In other words, the existence of a fixed zone—whether that zone is large or small in size—breaks initial symmetry in the specific manner: the direction of elongation is perpendicular to the fixed zone. On the contrary, all tube shapes tend to converge to circular outlines when no fixed zone exists. Morita (1991b), Morita (1993) and Morita (2003) argues that this fixed zone represents whorl overlap and may explain why most open coiling or minimally overlapping gastropods have circular apertures. On the other hand, outer apertural lips accompanied by a distinct whorl overlap zone are either extended perpendicularly to the overlap zone or are abapically inflated.

In Plectostoma concinnum, there are extensive shape differences between the spire and the tuba apertures- notably the part of the aperture which was previously in contact with the previous whorl exhibits smoothed corners in the open-coiling phase and is more symmetrical than before (Fig. 5B). However, the aperture shape of the tuba is not tending towards a circle but has an ovate shape that is elongated along the anteroposterior axis where the ribs are forming. Morita did not address the case of ornamented specimens, so our data is not well suited to test the predictions of this model in its current state.

Number of times and trend in the switching between whorl growing mode and rib growing mode

The total number of ribs (i.e., number of switches between whorl growing and rib growing mode) can vary substantially between individuals even if they are of similar shell size (i.e., similar ontogeny axis length). The number of switches between these two growing modes also does not affect the final ontogeny axis length. However, we could not determine whether a shell with dense ribs would need more time to become fully grown and whether rib density differences would be related to the differences in the aperture size ontogeny profiles.

Despite differences in rib number, the trends in rib spacing patterns are similar. At the spire part, the spacing between ribs initially increases and then decreases towards where the tuba starts to detach from the spire (ca. 9–10 mm along the ontogeny axis, Fig. 1G). After that, the rib spacing increases again and reaches its maximum at the constriction (ca. 13–14 mm along the ontogeny axis).

A previous growth study on Plectostoma retrovertens showed that each rib represents a day of growth (Berry, 1962). However, Plectostoma concinnum ribs are heavier than those of Plectostoma retrovertens, and its commarginal ribs do not represent daily growth stages. Furthermore, our specimens were in a cohort and collected over the same period (i.e., under similar weather condition), thus the rib spacing pattern is unlikely to be caused by environmental factors.

When the trend between rib spacing (Fig. 8), aperture perimeters (Fig. 5A) and growth rates (Fig. 4) are examined closely, interesting relationships among these shell parameters emerge. First, the spacing between ribs is the largest, the aperture perimeter is the smallest, and the growth rate is the highest at the constriction phase. Second, the rib spacing increases together with the increase of the growth rate along the spire ontogeny, while rib spacing decreases with decreasing growth rates in the tuba part. This suggests there might be a possible positive association between growth rate and rib spacing, and hence rib density. Further studies are needed to investigate whether this association is incidental or not. With limited data, we cannot decipher the ontogenetic mechanisms that produce the ribs. Although several theoretical mechanisms have been proposed (e.g., Hammer, 2000; Moulton, Goriely & Chirat, 2012; Chirat, Moulton & Goriely, 2013), the actual biological processes responsible for the growth of commarginal ribs remain poorly understood. Hence, we suggest that future studies examine the growth rates (shell deposition rate) in relation to ornamental patterns to improve our understanding of the possible relationship between rib frequency and growth rates.

Conclusion

In this study, we have developed an approach which can be used to extract aperture morphological changes along the ontogeny from a shell and we have found a way to analyse the growth and form parameters simultaneously. By analysing growth and form in this irregularly coiled shell, we have shown the associations between aperture ontogeny profiles and growth rate in the determination of final shell form. Our aperture ontogeny profile analysis of the shell and observations on living specimens provide for the first time direct evidence for the mechanism behind the irregularly coiled shell: the rotational changes of animal and mantle edge during the shell ontogeny. Overall, we have also highlighted that there is a need to improve our understanding of the developmental biology of snails, especially with reference to the mantle and columellar muscular systems and their potential relationship to shell morphogenesis.

Although our study provides little direct information on the developmental and genetic factors that govern the shell growth and form, it already highlights some plausible constraints—related to the columelar muscle and living position—underlying the three shell ontogeny phases and two different growth modes of this species. As these three phases are known to occur in all of the species in this genus, including those with more regularly coiled shells, our results may be generalised further in the future. Our study sets the stage for future studies using mollusc species in general to address issues concerning the ecology, the evolution and the development of mollusc using a mixture of insights coming from aperture ontogeny profiles obtained by a 3D morphometric approach.

Supplemental Information

File S1 Microclimatic variation for Pangi plots

Click here for additional data file.

File S2 Raw image of 65 measured shells for Part 1 and Part 3(b)

Click here for additional data file.

File S3 Correlation between 2D and 3D arc length measurement of three specimens

Click here for additional data file.

File S4 Raw data for analysis in Parts 1, 2 and 3, and the R scripts

Click here for additional data file.

File S5 Python scripts for 3D aperture morphometrics and growth trajectory analysis

Click here for additional data file.

File S6 Digitalised aperture outlines (n = 33) of 5 specimens used in Part 2(a)

Click here for additional data file.

File S7 Comparison between raw digitised and Elliptical Fourier reconstructed aperture outlines

Click here for additional data file.

File S8 Data of rotation analysis in Blender format for Part 2(b)

Click here for additional data file.

File S9 Digitalised 3D ontogeny axis of a shell for torsion and curvature analysis in Blender

Click here for additional data file.

File S10 Absolute shell whorl arc length added during the growth experiments

Click here for additional data file.

File S11 PCA results of Elliptical Fourier coefficient for aperture shape analysis in Part 2(a)

Click here for additional data file.

We wish to thank Asni and Harizah’s family from Kampung Sukau for logistic support in the field. We are indebted to Willem Renema (Naturalis) for help in CT-scanning. We are also grateful to Reuben Clements and Heike Kappes for comments on the early version of this paper.

Additional Information and Declarations

Competing Interests

Author Contributions

Field Study Permissions

Data Deposition

Thor-Seng Liew, Annebelle C.M. Kok, Menno Schilthuizen and Severine Urdy are employees of the Naturalis Biodiversity Center; Severine Urdy is an employee of The Centrum Wiskunde & Informatica. The authors have declared that no competing interests exist.

Thor-Seng Liew conceived and designed the experiments, performed the experiments, analyzed the data, contributed reagents/materials/analysis tools, wrote the paper, prepared figures and/or tables.

Annebelle C.M. Kok conceived and designed the experiments, performed the experiments, analyzed the data, contributed reagents/materials/analysis tools, reviewed drafts of the paper, prepared figures and/or tables.

Menno Schilthuizen conceived and designed the experiments, contributed reagents/materials/analysis tools, reviewed drafts of the paper.

Severine Urdy conceived and designed the experiments, analyzed the data, contributed reagents/materials/analysis tools, wrote the paper.

The following information was supplied relating to ethical approvals (i.e., approving body and any reference numbers):

This work was carried out under the research permit from Economic Planning Unit, Malaysia (UPE: 40/200/19/2524) and permission from the Wildlife Department and the Forestry Department of Sabah.

The following information was supplied regarding the deposition of related data:

http://dx.doi.org/10.6084/m9.figshare.960018.

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
