# Peer review of "On growth and form of irregular coiled-shell of a terrestrial snail: Plectostoma concinnum (Fulton, 1901) (Mollusca: Caenogastropoda: Diplommatinidae)"

_PeerJ, doi:10.7717/peerj.383_

## Round 0.1 · original submission · Minor Revisions

Two reviews are now in hand, and both recommend acceptance with relatively minor revisions. Both reviewers make helpful suggestions that will improve your study, and they urge that some of your statements be phrased more carefully or conservatively (e.g., the impact of small sample sizes; better/more citations).

·

Basic reporting

The authors make several statements in the introduction which are not necessarily all broadly accepted or known, which need to be back-up (better) by citing some relevant references. Several statements are made without citing important or relevant literature (see my comments to the authors in the pdf)

The authors should consider not only discussing similarities between heteromorph gastropods and ammonoids, but also potential differences in ecology and growth and how they might be relevant for their interpretations. The term "heteromorph" should be more clearly defined for gastropods (and other shelled molluscs) as from time to time, the reader might not be aware that the term might differ between different groups and that the authors are sometimes discussing ammonoids too (see comments in pdf)

Experimental design

The experimental design seems to be in order as far as i could verify. I missed a small word on how similar the environmental conditions of the selected rock surfaces had to be to be selected.

Validity of the findings

It would be better to use the spearman (rank) correlation for all the correlations (even the normally distributed ones) as it is more suitable for monotonic (including non-linear and linear) functions and directly comparable with one another.

Additional comments

Some additional remarks can be found in the pdf

·

Basic reporting

The article is well-written and coherently structured, with sufficient introduction and background.

Experimental design

The research question is clearly defined, and the research conducted in a thorough and rigorous manner to high technical standard.

Validity of the findings

The findings are generally well supported. However, I do see a few issues in this regard:

1. Fig 4, at a glance, seems to demonstrate very little; that is visually the data do not seem to show any correlation between ontogeny axis and growth rate. I understand that statistically, you can find a correlation. But, given that the data point with the smallest growth rate is at the point where you argue the growth rate to be highest, I wonder whether more tempered, less definitive statements should be used. For instance, “These data suggest that O. concinnum may follow a S-shaped growth curve” seems more appropriate than “These data demonstrate that O. concinnum follows a S-shaped growth curve”
2. Fig 5A potentially shows an interesting trend: not so much that the aperture perimeter increases linearly, which I don’t find surprising, but that the rate of increase would be the same for different specimens. However, given the very small sample size, I don’t think this actually can be claimed. In fact, comparing specimen GK1-88 and GK1-94, GK1-88 is increasing at a lower rate. Hence, I think the small sample size should be addressed and the rate that is given on line 409 can at best be understood as an average.
3. You mention that you find no relation between the number of ribs and shell size, as characterised by ontogeny axis length. However, I suspect that aperture size must play an important role as well (n.b. comment 2). If you followed two shells with the same ontogeny axis length and the same aperture size/shape (and thus in the same growth stage), would you see the same rib frequency? This seems an important aspect, and one that probably cannot properly be addressed without a larger sample size of aperture size. Without such an investigation though, I’m not sure the statement that ‘the number of ribs varies substantially between individuals of similar shell size’ (line 591) is well supported.

Additional comments

This is a very useful study for understanding shell growth and form. It is thorough and well-conducted, the type of study that hopefully becomes more common. It also provides a firm basis on which to connect to theoretical models of shell form. For instance, within the spire stage, the growth is very regular, with well-defined aperture increase and mantle rotation. Such rates map directly to a growth model we developed [1], and it is very interesting to be able to see quantifiable rates that can be fed into such a model.


1. Moulton, D. E., & Goriely, A. (2012). Surface growth kinematics via local curve evolution. Journal of Mathematical Biology, 68(1-2), 81–108.

---

## Round 0.2 · accepted · Accept

Thank you for your careful and thoughtful responses to issues and questions raised by the 2 reviewers. I believe your study is now acceptable for publication, and that is my formal recommendation. Congratulations.